# Effect of Fe Addition on Microstructure and Mechanical Properties of As-cast Ti_49_Ni_51_ Alloy

**DOI:** 10.3390/ma12193114

**Published:** 2019-09-24

**Authors:** Peiyou Li, Yuefei Jia, Yongshan Wang, Qing Li, Fanying Meng, Zhirong He

**Affiliations:** 1School of Materials Science and Engineering, Shaanxi University of Technology, Hanzhong 723001, China; hanwangyongshan@163.com (Y.W.); liqing20190208@163.com (Q.L.); aboying@yahoo.com (F.M.); hezhirong01@163.com (Z.H.); 2School of Materials Science and Engineering, Shanghai University, Shanghai 200444, China; YuefeiJia@shu.edu.cn

**Keywords:** Ti–Ni–Fe alloys, shape memory alloy, mechanical properties, microstructures

## Abstract

Effect of Fe addition on microstructure and mechanical properties of as-cast Ti_49_Ni_51_ alloy were investigated. The experimental results shows the microstructures of Ti_48.5_Ni_51_Fe_0.5_ and Ti_48_Ni_51_Fe_1_ alloys are mainly composed of TiNi matrix phase (body-centered cubic, BCC), Ti_3_Ni_4_ and Ni_2.67_Ti_1.33_ phases; the microstructure of Ti_47_Ni_51_Fe_2_ alloy is mainly composed of BCC TiNi, Ti_3_Ni_4_, Ni_2.67_Ti_1.33_, and Ni_3_Ti phases; the microstructure of the Ti_45_Ni_51_Fe_4_ alloy is mainly composed of TiNi, Ti_3_Ni_4_ and Ni_3_Ti phases. The Ni_3_Ti nanocrystalline precipitates at the adjacent position of Ni_2.67_Ti_1.33_ phase. The Ti_48.5_Ni_51_Fe_0.5_ and Ti_48_Ni_51_Fe_1_ alloys have high yield strength and fracture strength, and can be as the engineering materials with excellent mechanical properties. In addition, the Ti_48.5_Ni_51_Fe_0.5_ alloy with the low elastic modulus and large elastic energy is also a good biomedical alloy of hard tissue implants. The fracture mechanism of the four alloys is mainly cleavage fracture or quasi-cleavage fracture, supplemented by ductile fracture. The experimental data obtained provide the valuable references in application of as-cast alloys and heat-treated samples in the future.

## 1. Introduction

Ti–Ni-based shape memory alloys (SMAs) with near or equal atomic ratios are favored by material researchers for their unique shape memory effect, excellent superelasticity, good mechanical properties, good corrosion resistance and elastocaloric effect [1,2,3,4]. In recent years, Ti–Ni-based alloys with excellent properties have been widely used in the fields of biomedicine, microelectromechanical systems, aviation and solid-state cooling [1,2,3,4,5,6]. In the field of biomedical alloys, Ti–Ni-based alloys are mainly used as hard tissue and soft tissue implants, as well as medical devices [2,6]. In microelectromechanical systems, the functional microgrippers, fluid-flow valves, and micromirror actuators are fabricated by using the Ti–Ni-based alloys [7,8]. In aeronautical and space devices, Ti–Ni-based alloys are mainly used as cryogenic fittings in hydraulic systems of military aircraft, and the blade trajectory controllers of helicopter [5]. In addition, the elastocaloric effect of SMAs has been successfully exploited in solid-state refrigeration as an alternative to vapor-compression-based cooling [9]. In fact, the microstructures and transformation deformation of TiNi alloys used in different fields are also different. Addition of elements, casting process and heat treatment can change the microstructure and phase transformation behavior of TiNi alloys [10,11,12,13,14,15]. To meet the application in different fields, the main additive elements are Pb, Cu, Hf, Nb, Co, and Fe [10−14]. In ternary Ti–Ni-based alloys, Ti–Ni–Fe alloys have been widely used for pipe joints owing to their excellent shape memory effect [16,17,18]. 

Among the reported ternary Ti–Ni–Fe alloys, it is reported that Ti-rich Ti–Ni–Fe alloys are obtained by substituting minor Fe elements for Ni elements in isoatomic TiNi alloys. In early studies, the cold-rolled Ti_50_Ni_47.5_Fe_2.5_ (at.%) alloy has two-way shape memory effect and low martensite transformation temperature [19]. Due to the excellent shape memory effect of Ti-rich Ti–Ni–Fe alloys, some researchers have developed the different Ti-rich Ti–Ni–Fe alloys [18,19,20,21,22,23,24,25,26,27,28]. The shape memory effect of Ti_50_Ni_47_Fe_2.5_Nd_0.5_ alloy is improved when the trace amount of Nd is substituted for Ni [20], comparing with the Ti_50_Ni_47_Fe_2_Mo_1_ and Ti_50_Ni_48_Fe_2_ alloys [21,22]. In addition, the martensitic transformation of Ti_50_Ni_48_Fe_2_ alloy treated by swaging at the different temperatures is hindered by dislocation introduced by swaging [23]. After swaging, a small and wide B2→R peak can be observed, which seriously hinders the R→B19′ transformation [23]. Nagase reported the amorphization tendency of a B2 phase in Ti_50_Ni_48_Fe_2_ and Ti_50_Ni_10_Fe_40_ intermetallic compounds to determine the relationship between solid-state amorphization and martensite transformation; the condition of solid-state amorphization observed is at 103 K and 298 K by high-voltage electron microscopy [24]. In fact, the Ti_50_Ni_48_Fe_2_ alloy can maintain stable phase transition temperature after 50 thermal cycles [25]. In recent years, the microstructures of hot deformed Ti_50_Ni_47_Fe_3_ alloy reported are strain localization, grain boundary serrations, and relatively fine near-equiaxed grains; at higher working temperature and higher strain rate, the content of dynamic recrystallization is higher; the increase of dynamically recrystallized grains results in the restraint of austenitic ↔ martensitic transformation [26,27]. Liang et al. [28] reported that the Ti_51.8_Ni_45_Fe_3.2_ alloy comprises a B2 austenite matrix and a Ti_2_Ni precipitate, and in the Ti_51.8_Ni_44_Fe_3.2_Nb_1_ alloy with high yield strength, the Ti_2_Ni and β-Nb precipitates occur in the matrix of B2 austenite. In the Ti-rich Ti–Ni–Fe alloys, to obtain a large number of B2 TiNi phases, the heat-treated ingots were at high temperature for a long time, and the samples were cooled rapidly in ice water after heat treatment [18,19,20,21,22,23,24,25,26,27,28]. After heat treatment, some researchers have carried out subsequent high temperature forging, or low-temperature aging treatment, or thermal cycle to further study the relationship between microstructure, phase transformation and mechanical properties [18,19,20,21,22,23,24,25,26,27,28,29]. As an alloy system, to study the microstructures and mechanical properties of Ti–Ni–Fe alloy system under different conditions more systematically, little research has been done on the microstructures and mechanical properties of ingots with a given Ti–Ni–Fe alloy system. The study of the microstructures and mechanical properties of ingots is also to better understand the evolution of the microstructures and phase transformation in the subsequent high-temperature heat-treated and low-temperature aging samples. According to the reported references we know, the Ti-rich Ti–Ni–Fe alloys have been reported extensively, while the Ni-rich Ti–Ni–Fe alloys are rarely reported. 

The Ti_49_Ni_51_ alloy is a shape memory alloy with excellent properties because of its excellent superelasticity under medium temperature aging [30]. Therefore, in this paper, the Ni-rich Ti_49_Ni_51_ alloy is used as the base alloy. The Ni-rich Ti–Ni–Fe alloys are obtained by substituting trace Fe element for Ti element. The microstructure, phase composition and mechanical properties of the Ti–Ni–Fe ingots prepared by vacuum arc meltingare were investigated by using X-ray diffractometer (XRD), optical microscope (OM), scanning electron microscopy (SEM), transmission electron microscope (TEM), high resolution transmission electron microscope (HRTEM), and electronic testing machine. The experimental data obtained provide the valuable references on the microstructural evolution, phase transformation or mechanical properties for subsequent heat treatment at high temperature, aging treatment at medium and low temperature, or other conditions in the future.

## 2. Experimental Procedure

The combinations of pure Ti, Ni and Fe (purity of 99.9% or higher) were used to prepare ingots under a high vacuum (3 × 10^−3^ Pa) using a non-consumable arc-melting furnace (Shenyang Scientific Instruments Co., Ltd. of Chinese Academy of Sciences, Shenyang, China) in an argon atmosphere. The nominal compositions of alloys are Ti_48.5_Ni_51_Fe_0.5_, Ti_48_Ni_51_Fe_1_, Ti_47_Ni_51_Fe_2_, and Ti_45_Ni_51_Fe_4_ (at.%). After mixing the raw materials, a total of 20 g of raw materials were put into a water-cooled copper crucible. To ensure the uniformity of the chemical composition, the ingots were smelted more than 4 times. The samples for mechanical and structural analysis were cut from the alloy ingots by using a slow steel saw and electric discharge wire-cutting technology. To measure the phase of the alloy more accurately, three 3 mm thin sheets were placed on a plane for XRD diffraction (Rigaku Company, Tokyo, Japan), at an operating voltage of 30 kV, using Cu-Kα radiation. The surfaces of thin sheets with diameter of 3 mm were polished using standard metallographic procedures, consisting of grinding up to 2000 grit with SiC paper and polishing with a colloidal silica suspension. For observing the microstructure, the surfaces of polished samples were etched. The etched solution is a mixed solution of HF, HNO_3_, and H_2_O, and the ratio of the corresponding volumes is 1:4.5:4.5. The microstructures of the etched samples were observed by inverted OM (Shanghai Changfang Optical Instrument Co., Ltd., Shanghai, China).

The mechanical properties of the samples were tested by quasi-static compression at room temperature. According to the standard compressed testing process (GB/T 7314–2005 standard of China), the ratio of length (*L*) to diameter (*d*) is in the range of 1 to 2. The cylindrical Ti–Ni–Fe samples with diameter of 3.0 mm and length of ~5.0 mm were tested under quasi-static uniaxial compression. The testing machine is a CMT5105 electronic testing machine (Metis Industrial Systems (China) Co., Ltd., Shanghai, China). Alloys for each component were measured for three times at a strain rate of 2.5 × 10^−4^ s^−1^, and the reported experimental data are the average of three measurements. After the compression, the surfaces morphology were observed using SEM (JSM 6390LV, Metis Industrial Systems (China) Co., Ltd., Shanghai, China). TEM equipped with an energy dispersive spectroscope (EDS) analyzer and HRTEM observations were carried out on a JEM–2100F microscope (JEO Road (Beijing) Science and Trade Co., Ltd. Shanghai Branch, Shanghai, China) operating at 200 kV. The thin specimens for TEM observation were prepared by double-jet electrolysis thinning method in an electrolyte solution of 25% HNO_3_ and 75% methanol by volume around 238 K.

## 3. Results and Discussion

### 3.1. Microstructure of Ti–Ni–Fe Alloys

Figure 1 shows the XRD patterns of Ti–Ni–Fe alloys. The microstructures of Ti_48.5_Ni_51_Fe_0.5_ and Ti_48_Ni_51_Fe_1_ alloys are mainly composed of TiNi matrix phase (body-centered cubic, BCC), Ti_3_Ni_4_ phase (trigonal) and Ni_2.67_Ti_1.33_ phase (trigonal). The microstructures of Ti_47_Ni_51_Fe_2_ alloy are mainly composed of BCC TiNi, Ti_3_Ni_4_, Ni_2.67_Ti_1.33_, and Ni_3_Ti phases (close-packed hexagonal). For the Ti_47_Ni_51_Fe_2_ alloy, the more diffraction peaks of the second Ni_3_Ti phase were observed in Figure 1a, and the large intensity of the diffraction peaks indicates the larger content of the second Ni_3_Ti phase. For the Ti_45_Ni_51_Fe_4_ alloy, the microstructure is mainly composed of TiNi, Ti_3_Ni_4_ and Ni_3_Ti phases, and the diffraction intensity of the strongest diffraction peaks of Ti_3_Ni_4_ and Ni_3_Ti phases is larger, comparing to the remaining three alloys, which indicates the large contents of the two phases. Since the four alloys are as-cast alloys, the orientation of each grain is almost completely disorder, that is to say, the orientation probability of the grain is the same, or textures are random. Therefore, the intensity of XRD diffraction peak can be used to quantify phase fraction. In fact, the intensity of the diffraction peaks of each phase is approximately proportional to the volume fraction and content [31,32]; based on this relation, the volume fraction (*V*_i_) of the corresponding phase can be estimated according to the relative diffraction intensity of the strongest diffraction peaks of the phase [32]. In Figure 1, the diffraction intensity of TiNi phase with the (110) plane is normalized to 100%, and the diffraction intensity of the strongest diffraction peaks of other phases is normalized. The normalized value of each phase is approximately equal to the relative volume fraction (*V*_ri_) of corresponding phase. The relationship between the *V*_r*i*_ values of the corresponding phases and the approximate volume fraction (*V*_1_) of the BCC TiNi phase is [32]: (1)$$\sum_{i=1}^{n}V_{1}V_{ri}=100\%\;(i\;=\;1,\;2,\;3,\ldots ,\;n)$$
*V_i_* = *V*_1_*V*_r*i*_ (*i* = 1, 2, 3,…, *n*)(2)

*V_i_* is the approximate volume fraction of the *i* phase. The approximate volume fractions of the phase calculated are listed in Table 1. For the Ti_48.5_Ni_51_Fe_0.5_, Ti_48_Ni_51_Fe_1_ and Ti_45_Ni_51_Fe_4_ alloys, the volume fractions of TiNi phase are 74.24%, 72.32% and 74.26%, respectively. The difference of volume fraction of the three alloys is small. However, the volume fraction of TiNi phase in Ti_47_Ni_51_Fe_2_ alloy is 68.49%, which is smaller than those in the other three alloys. In fact, the addition of Fe has the small effect on the volume fraction of TiNi phase. In previous studies, the phase of as-cast Ti_49_Ni_51_ alloy mainly consists of B2 TiNi matrix phase, the Ti_3_Ni_4_ and Ti_2_Ni second phases. The addition of Fe changes the matrix phase of Ti_49_Ni_51_ alloy. The diffraction peaks of B2 TiNi phase were not detected in Figure 1. The addition of Fe also changes the types and contents of the second phase. For the Ti_48.5_Ni_51_Fe_0.5_ alloy, the volume fraction of Ti_3_Ni_4_ phase is 16.41%, which is larger than that of the corresponding phases of the other three alloys. For the Ti_48_Ni_51_Fe_1_, Ti_47_Ni_51_Fe_2_ and Ti_45_Ni_51_Fe_4_ alloys, the volume fractions of Ti_3_Ni_4_ phase increase with the increase of Fe content. For the Ti_48_Ni_51_Fe_1_ alloy, the volume fraction of Ni_2.67_Ti_1.33_ phase with the large content is 12.81%, while for the Ti_48.5_Ni_51_Fe_0.5_ and Ti_47_Ni_51_Fe_2_ alloys, the contents of the Ni_2.67_Ti_1.33_ phase are approximately the same. In addition, for Ti_45_Ni_51_Fe_4_ alloy, no diffraction peaks of corresponding Ni_2.67_Ti_1.33_ phase were found in XRD spectrum. These results indicate that the addition of Fe has a great influence on Ni_2.67_Ti_1.33_ precipitation. For the Ti_48.5_Ni_51_Fe_0.5_ and Ti_48_Ni_51_Fe_1_ alloys, the diffraction peaks of Ni_3_Ti phase were found in Figure 1, but for the Ti_47_Ni_51_Fe_2_ and Ti_45_Fe_51_Fe_4_ alloys, the volume fractions of Ni_3_Ti phase increase with the increase of Fe content. Therefore, with the increase of Fe content, and the decrease of corresponding Ti content, the atomic ratio of Ni to Ti increases, which leads to the precipitation of Ni_3_Ti intermetallic compound with stable high melting point, in the cooling process of alloy solution.

Figure 2 shows the microstructure of Ti–Ni–Fe alloys. In Figure 2a, except for the matrix, the second phase presents two shapes, one with the more particles is elliptical and quadrilateral, and the other with the less particles is fibrous. According to XRD quantitative analysis, for the Ti_48.5_Ni_51_Fe_0.5_ alloy, the content of Ti_3_Ni_4_ phase is larger than that of Ni_2.67_Ti_1.33_ phase. It can be judged that the elliptical or quadrilateral second phase is the Ti_3_Ni_4_ phase, while the fibrous particles are the Ni_2.67_Ti_1.33_ phase. Figure 2a shows the precipitation of the Ti_3_Ni_4_ phase in TiNi matrix. The maximum particle size of Ti_3_Ni_4_ phase is 10 μm, and the minimum particle size is 2 μm. With the increase of Fe content and the corresponding decrease of Ti content, for the Ti_48_Ni_51_Fe_1_ alloy, the content of Ti_3_Ni_4_ phase decreases obviously, and most of the particle size decreases; the Ni_2.67_Ti_1.33_ phase with fibrous shape increases obviously, and most of the fibrous particles aggregate into clusters, as shown in Figure 2b. For the Ti_47_Ni_51_Fe_2_ alloy, the content of fibrous Ni_2.67_Ti_1.33_ phase decreases significantly, and most of the fibrous Ni_2.67_Ti_1.33_ phase still aggregates into clusters in Figure 2c, compared with the Ti_48_Ni_51_Fe_1_ alloy; the Ti_3_Ni_4_ phase is still presented as elliptical granules; besides the particles of these two phases, according to XRD analysis, the large area of white particles are Ni_3_Ti phase. For the Ti_45_Ni_51_Fe_4_ alloy, the content of fibrous Ni_2.67_Ti_1.33_ phase is the lowest in Figure 2, and the length of fibers and the size of clusters are also significantly reduced. As the low content of Ni_2.67_Ti_1.33_ phase, no diffraction peak was found in XRD pattern. In fact, the content of Ti_3_Ni_4_ second phase in Ti_45_Ni_51_Fe_4_ alloy is larger than that of Ti_47_Ni_51_Fe_2_ alloy in Figure 2d, which is consistent with the data calculated by XRD. With the increase of Fe content, the Ti_3_Ni_4_ phase is a stable second phase, and keeps a high content in the four alloys; the variation of content of Ni_2.67_Ti_1.33_ phase is large, and the variation of size of fiber clusters also is large, which shows that the stability of Ni_2.67_Ti_1.33_ phase is weaker than that of Ti_3_Ni_4_ phase. In binary Ti–Ni alloys, the Ni_3_Ti phase is a stable intermetallic compound with high melting point. Accordingly, for the Ni-rich Ti_47_Ni_51_Fe_2_ and Ti_45_Ni_51_Fe_4_ alloys, the Ni_3_Ti phase can precipitate stably; when the more Ni_3_Ti phase precipitates, the content of Ni_2.67_Ti_1.33_ phase decreases, suggesting that the stability of Ni_3_Ti phase is stronger than that of Ni_2.67_Ti_1.33_ phase.

To further study the microstructures of the alloy, the EDS measurements were carried out in Ti_48.5_Ni_51_Fe_0.5_ alloy. Figure 3a is a TEM scan image. In Figure 3a, the block second phase is labeled Point 1, the short fibrous second phase is labeled Point 2, and the matrix is labeled Point 3. The EDS point scan is performed at the labeled position, and the contents of Ti, Ni and Fe elements obtained are listed in Table 2. The Ti, Ni and Fe contents at Point 1 were 41.3 at.%, 57.9 at.% and 0.9 at.%, respectively. The atomic ratio of Ti to Ni is 1:1.4, which is close to the atomic ratio of Ti to Ni in Ti_3_Ni_4_ phase. It shows that the particle with quadrilateral shape is the Ti_3_Ni_4_ phase, which is consistent with the analysis results in Figure 2a. The Ti and Ni contents at Point 2 are 34.2 at.% and 65 at.%, respectively, and the atomic ratio of Ti to Ni is 1:1.9, which is very close to the atomic ratio of Ni_2.66_Ti_1.33_ phase. It shows that Ni_2.66_Ti_1.33_ phase presents fibrous particles, which is consistent with the analysis of the microstructure in Figure 2. The atomic ratio of Ti to Ni at Point 3 is 1.03:1, which is close to the atomic ratio of TiNi phase, indicating that the matrix is TiNi phase, which is consistent with the XRD analysis results. In addition, the Fe content in the matrix is larger than that in the second phase, which is related to the heat of mixing of Ti–Fe and Ni–Fe atomic pairs. The heat of mixing between atomic pairs can indirectly react with the interaction force between atoms pairs [31,32,33,34]. When the heat of mixing of atomic pairs is negative, the interaction force between atomic pairs is large, when the heat of mixing of atomic pairs is small negative or positive, the interaction force between atomic pairs is weak [31,34]. The heat of mixing of Ti–Fe atom pair is −17 kJ⋅mol^−1^ [33], which is larger than that of Ni–Fe atom pair (−2 kJ⋅mol^−1^ [33]), indicating that the interaction force of Ti–Fe atom pair is larger than that of Ni–Fe atom pair. In the matrix, the content of Ti atom is larger than that of Ti atom in the second phase, and the larger Ti content promotes the aggregation of the more Fe atom and Ti atoms. In the second phase, the content of Ni atom is larger than that of Ni atom in the matrix, while the content of Ti atom is lower than that of Ti atom in the matrix. As a result, the Fe atom with less content is retained in the second phase, and some of the Fe atoms diffuse towards the Ti-rich region of matrix. In fact, in Table 2, it can be found that the change of Ti content is consistent with that of Fe content, while the change of Ni content is contrary to that of Fe content. In addition, the content of Fe is obviously larger than that of Fe in the nominal composition (0.5 at.%) of the alloy, and the specific reasons need to be further investigated.

Figure 4 shows that the TEM bright field images of Ti_48.5_Ni_51_Fe_0.5_ alloy. In Figure 4a, the circular particles are precipitated in the matrix, and the ranging in size is from 100 to 200 nm, while the elliptical particle is larger than 300 nm. According to the analysis in Figure 3, these particles are Ti_3_Ni_4_ phase. In Figure 4a, for a large number of strip-like particles, its length ranges from 200 nm to 1 μm, according to the EDS analysis, the strip-like particles are the Ni_2.67_Ti_1.33_ phase. In Figure 4d, the content of Ti_3_Ni_4_ phase in circle, ellipse and polygon is obviously larger than that in strip or fibrous Ni_2.67_Ti_1.33_ phase, which is consistent with the XRD results. The particle size of Ti_3_Ni_4_ phase is mostly several hundred nanometers, while the particle length of Ni_2.67_Ti_1.33_ phase is mostly less than 500 nanometers. A large number of nano-sized particles precipitated in the matrix phase may inhibit the formation of B2 phase, and promotes the BCC TiNi phase in Ti–Ni–Fe alloys. To better determine the microstructure of the matrix phase, the corresponding selected area electron diffraction (SAED) of the matrix marked as the “A” area in Figure 4a was carried out, and the diffraction spots are shown in Figure 4b. The analysis of diffraction spots can know that the matrix TiNi phase is a body-centered cubic structure with [113] crystal band axis. Three typical (110), (211) and (121) crystal planes are also marked on Figure 4b. In Figure 4a, the diffraction spots of SAED for the particle labeled as “B” are shown in Figure 4c. The crystal band axis [001] of the Ti_3_Ni_4_ phase with a tripartite structure is exhibited in Figure 4c, and three (110), (320) and (230) crystal planes are labeled in Figure 4c, respectively. When the fiber-like particles labeled C in Figure 4e are selected for electron diffraction, the diffraction spots of BCC TiNi and Ni_2.67_Ti_1.33_ phases are shown in Figure 4e. Because of the small size of the fibrous particles, it is easy to include the diffraction spots of the matrix phase in the electron diffraction. Through analysis, the crystal zone axis of TiNi phase is [102], the crystal planes are (020), (211) and (211). In addition, the crystal zone axis [821], and the (108), (1110) and (012) crystal planes of the Ni_2.67_Ti_1.33_ phase are marked in Figure 4e, respectively.

In Figure 4e, in addition to two sets of diffraction spots of TiNi and Ni_2.67_Ti_1.33_ phases, the dimmer diffraction spots marked by white light are also found. These dimmer diffraction spots indicate the presence of nanoparticles containing another phase near or inside the fibrous particles containing the Ni_2.67_Ti_1.33_ phase. Because of the small number of dark diffraction spots, the difficulty of calibration is increased. To better observe the nanoparticles of another phase near or inside the Ni_2.67_Ti_1.33_ phase, the fibrous particles labeled C in Figure 4d were observed by high resolution transmission electron microscope (HRTEM), and the HRTEM image is shown in Figure 4f. By Fourier transform, the diffraction image of nanocrystals located at red square at the left in Figure 4f is shown in Figure 4g. Based on diffraction spots analysis, the microstructure at the red area on the left side of Figure 4f is the Ni_2.67_Ti_1.33_ phase, and the crystalline zone axis [821], the (1110) and (012) crystal planes are labeled in Figure 4g, respectively. When Fourier transform is applied to the red block diagram on the right side of Figure 4f, the diffraction image is shown in Figure 4h. According to calibration of diffraction spots, the crystalline band axis of Ni_3_Ti phase is [010], and the crystal planes are (201), (203) and (004). The results in Figure 4f suggest that the Ni_3_Ti nanocrystals precipitates at the adjacent position of Ni_2.67_Ti_1.33_ phase.

### 3.2. Mechanical Properties of Ti–Ni–Fe Alloys

Figure 5 shows the nominal stress–strain curves of Ti–Ni–Fe alloys. Since each alloy is tested three times, each alloy has three stress–strain curves; in Figure 5, the experimental data on the selected stress–strain curves are close to the average of the three measurements. For the as-cast Ti_49_Ni_51_ alloy, with the increase of stress, B2 austenite phase can be transformed into martensite phase, showing a martensite transformation plateau in the stress–strain curves [30]. No martensite transformation plateau is found on the nominal stress–strain curve in Figure 5, because the matrix phases of the four alloys are not the B2 TiNi phase with the large grain sizes, but the TiNi phase with body-centered cubic structure. According to the reported results of Sun et al. [4,35,36,37], the strain of martensite transformation plateau decreases as the particle size of B2 phase decreases; when the particle size reaches 10 nm, the martensite transformation plateau approximately disappears, i.e., the stress–strain curve presents a smooth stress-induced phase transition. In Figure 5, the smooth elastic deformation may include continuous stress-induced phase transition, i.e., the phase transition of nanocrystals under stress, resulting in an increase in elastic strain [4,35,36,37]. Among the four alloys, the Ti_48.5_Ni_51_Fe_0.5_ alloy is the representative one. In Figure 5, the maximum elastic strain of Ti_48.5_Ni_51_Fe_0.5_ alloy reaches 2.88%, which is obviously larger than 2% of the structural material, and also larger than the elastic strain of the other three Ti–Ni–Fe alloys. The Ti_48.5_Ni_51_Fe_0.5_ alloy with 2.88% elastic deformation is a kind of smooth hardening superelasticity [4]. This kind of smooth hardening superelasticity is different from the pseudoelasticity of shape memory alloy [4,35,36,37]. The former originates from the elastic deformation of nanocrystalline microstructures, while the latter is the elastic deformation caused by martensitic transformation with larger particle size [4,35,36,37]. According to the results reported by Ahadi et al. [4], the deformation of nanocrystalline microstructures indicates that a small amount of B2 phase is transformed into B19′ phase, which also indicates that the current Ti_48.5_Ni_51_Fe_0.5_ alloy may contain a small amount of B2 nanocrystals, and the results need to be further studied in the future. The elastic strain of the other three alloys is close to 2%, which indicates that the ability of B2 nanocrystalline phase to transform into B19′ phase is weak, and the amount of B2 nanocrystalline in the alloys is small. In addition, the transition of nanocrystalline B2 phase to B19′ phase can produce elastocaloric effect in a wide temperature range [4]. Therefore, the elastocaloric effect of Ti_48.5_Ni_51_Fe_0.5_ alloy needs to be further studied in the future.

The mechanical data calculated from the stress–strain curves are listed in Table 1. The linear elastic limit (*σ*_e_) of alloys decreases from 1779 MPa of Ti_48.5_Ni_51_Fe_0.5_ alloy to 1463 MPa of Ti_45_Ni_51_Fe_4_ alloy. The yielding strength (*σ*_0.2_) of Ti_48.5_Ni_51_Fe_0.5_ and Ti_48_Ni_51_Fe_1_ alloys are 2066 and 2089 MPa, respectively, and larger than those of Ti_47_Ni_51_Fe_2_ and Ti_45_Ni_51_Fe_4_ alloys (1945 and 1865 MPa, respectively). The maximum compressive strength (*σ*_b_) of Ti_48_Ni_51_Fe_1_, Ti_47_Ni_51_Fe_2_ and Ti_45_Ni_51_Fe_4_ alloys are as large as 2500 MPa, while that of Ti_48.5_Ni_51_Fe_0.5_ alloy is as large as 2420 MPa. In Figure 5, as the difference between fracture strength and maximum compressive strength is small, the fracture strength can be approximately equal maximum compressive strength. The *σ*_0.2_ values of ternary Ti–Ni–Fe alloys are close to those of Ti–Cu–Ni–Zr bulk metallic glasses [38], while the *σ*_b_ values of ternary Ti–Ni–Fe alloys are larger than those of Ti–Cu-based bulk metallic glasses [38,39]. Therefore, due to the addition of Fe, the four alloys have high yield strength and fracture strength, and are an engineering material with excellent mechanical properties. When the strength is higher, the plasticity of the alloy is generally lower. The range of plastic strain (*ε*_p_) of the four alloys is from 2.2% to 3.0% in Table 1. Compared with the plastic deformation of as-cast Ti_49_Ni_51_ alloy, the addition of Fe reduces the plastic deformation of the alloy. In the linear elastic stage, the elastic modulus (*E*) of the alloy is equal to the ratio of stress to strain, that is, the slope of the linear elasticity. The calculated *E* values of four alloys are listed in Table 1. The *E* value of Ti_48.5_Ni_51_Fe_0.5_ alloy is 61.8 GPa, while those of Ti_48_Ni_51_Fe_1_ and Ti_45_Ni_51_Fe_4_ alloys are 80.2 GPa and 81.3 GPa, respectively. In addition, the *E* value of Ti_47_Ni_51_Fe_2_ alloy is 73.2 GPa. In fact, the elastic modulus of the four alloys is smaller than that of the commercial Ti-6Al-4V alloy (120 GPa) [40]. When the elastic modulus of implant is different from that of human tissue, the stress shielding is easy to occur, and the secondary injury is easy to occur to patients [40]. Accordingly, the Ti_48.5_Ni_51_Fe_0.5_ alloy with the low elastic modulus and high strength is a good biomedical alloy for hard tissue implants.

### 3.3. Elastic Energy and Toughness of Ti–Ni–Fe Alloys

As the elastic modulus of the four alloys are smaller than those of commercial medical alloys, they can be used as hard tissue implants of medical alloys. In application, the strength of the alloy is generally smaller than the linear elastic limit. When the alloy has the large elastic energies (*W*_e_), the alloy has high safety in application. Therefore, it is necessary to calculate the *W*_e_ values of four alloys. The elastic energy is equal to the area enclosed by the stress–strain curve and strain in the linear elastic stage, as shown in the shadow area of Figure 6. The Ti_48.5_Ni_51_Fe_0.5_ alloy has the largest *W*_e_ value (26.12 × 10^6^ J⋅m^−3^), which is larger than those of the other three alloys, and also larger than those of the Ti–11Nb–9Fe (7.1 × 10^6^ J⋅m^−3^) and Ti–6Al–4V (2.8 × 10^6^ J⋅m^−3^) biomedical materials [41,42]. In fact, the *W*_e_ values of the four alloys are larger than those of Ti–6Al–4V medical alloys. Therefore, the Ti–Ni–Fe alloys have high safety as hard tissue implants of medical alloys in application.

As the Ti–Ni–Fe alloys with the high strength can be used as engineering material, when the alloys have high compressive toughness, the safety of Ti–Ni–Fe alloys is high in the application. Figure 7 shows the quasi-static compressive toughness (*A*_t_) of Ti_48.5_Ni_51_Fe_0.5_ alloy. On the stress–strain curve, the toughness is equal to the area enclosed by the stress–strain curve and the strain, as shown in the shadow of Figure 7. When the alloy has high strength and low plastic deformation, the toughness of the alloy is smaller; when the alloy has low strength and large plastic deformation, the toughness of the alloy is smaller; high toughness requires that the alloy has medium strength and medium plastic deformation [32]. Using the area shown in Figure 7, the calculated *A*_t_ values of the four alloys are listed in Table 1. The *A*_t_ values of Ti_48.5_Ni_51_Fe_0.5_ and Ti_48_Ni_51_Fe_1_ alloys are 89.91 × 10^6^ J⋅m^−3^ and 85.23 × 10^6^ J⋅m^−3^, respectively, which are smaller than those of Ti_47_Ni_51_Fe_2_ (92.89 × 10^6^ J⋅m^−3^) and Ti_45_Ni_51_Fe_4_ (94.36 × 10^6^ J⋅m^−3^) alloys. In fact, the *A*_t_ values of the four alloys differ slightly. As the plastic deformation of the alloy is small, the *A*_t_ value is relatively small, compared with the reported Ti–Fe–Sn alloys [32]. In the future investigation, on the premise of maintaining the high strength of Ti–Ni–Fe alloy, the plastic deformation of the alloys needs to be improved, so as to improve the toughness of the alloys.

### 3.4. Fracture Morphology of Ti–Ni–Fe Alloys

Figure 8 shows the fracture morphology of the compressed samples. The fracture morphology of Ti_48.5_Ni_51_Fe_0.5_ alloy in Figure 8a shows a large area of cleavage plane and a small amount of river-like patterns, which indicates that the fracture mechanism of the alloy is cleavage fracture. In addition, a large area of crystal melting is found in Figure 8b. As a large amount of elastic energy is stored with the increase of stress in the region, the release of elastic energy leads to the increase of local temperature at the moment of specimen fracture. When the temperature of increase exceeds the melting point of the sample, a large area of crystal melting occurs on the fracture surface. In fact, due to the small plastic deformation of the alloy, the release of elastic energy during plastic transformation is less, which leads to the instantaneous release of accumulated elastic energy, and the local temperature rise results in the melting of the crystal. In the fracture morphology of bulk metallic glasses, a large number of melting phenomena of amorphous alloys caused by local temperature rise have been found [43]. Figure 8c,e are the fracture morphologies of the Ti_48_Ni_51_Fe_1_ and Ti_47_Ni_51_Fe_2_ alloys. In Figure 8c,e, in addition to the large area of cleavage plane, ductile fracture in a small area are also present. Figure 8d is a local enlargement of the ductile fracture region of Figure 8c. The transgranular fracture occurs at the second phase of larger particles, and the ductile tear region appears around the transgranular fracture zone. In Figure 8e, the transgranular fracture and ductile tearing are also observed. The ductile tearing region can absorb the plastic work during the formation process, so that the alloy has a plastic deformation. As the area of ductile tear zone is small, the plastic work absorbed is small, which leads to the small plastic deformation of the alloy. Figure 8g shows the fracture morphology of Ti_45_Ni_51_Fe_4_ alloy. Besides cleavage fracture, the quasi-cleavage fracture is shown in Figure 8g. In fact, the quasi-cleavage fracture and cleavage fracture are both characteristics of macro-brittle fracture. Figure 8h shows the ductile tear region and a small number of dimples. Both the ductile tear region and the formation of dimples can absorb plastic work, thus making the alloy have plastic deformation. In Figure 8g, the plastic strain of the alloy is small due to the small area of ductile tear zone, which is consistent with the result of Figure 5. Cleavage fracture generally occurs in the alloy with body-centered cube and dense hexagonal structure. The matrix of the four alloys is body-centered cubic, so the alloy is prone to cleavage fracture. When the area of cleavage fracture is large, the area of ductile fracture zone is relatively small, which results in less plastic work absorbed by ductile tear region, leading to the small plastic deformation of the alloy. Accordingly, the fracture mechanism of the four alloys is mainly cleavage fracture or quasi-cleavage fracture, supplemented by ductile fracture.

## 4. Conclusions

The microstructure and mechanical properties of rapidly solidified Ti–Ni–Fe alloys with high strength were investigated, and the main conclusion are as following.

(1)The microstructures of Ti_48.5_Ni_51_Fe_0.5_ and Ti_48_Ni_51_Fe_1_ alloys are mainly composed of BCC TiNi matrix phase, Ti_3_Ni_4_ and Ni_2.67_Ti_1.33_ phases. The microstructure of Ti_47_Ni_51_Fe_2_ alloy is mainly composed of BCC TiNi, Ti_3_Ni_4_, Ni_2.67_Ti_1.33_, and Ni_3_Ti phases. The microstructure of the Ti_45_Ni_51_Fe_4_ alloy is mainly composed of TiNi, Ti_3_Ni_4_ and Ni_3_Ti phases.(2)The Ni_3_Ti phase precipitates at the adjacent position of Ni_2.67_Ti_1.33_ phase. A large number of nano-sized particles precipitated in the matrix phase may inhibit the formation of B2 phase, and promotes the formation of BCC TiNi phase in Ti–Ni–Fe alloys.(3)The yielding strength of Ti_48.5_Ni_51_Fe_0.5_ and Ti_48_Ni_51_Fe_1_ alloys are 2066 and 2089 MPa, respectively; the two alloys with the high yield strength and fracture strength are an engineering material with excellent mechanical properties. The Ti_48.5_Ni_51_Fe_0.5_ alloy with the low elastic modulus and large elastic energy is a good biomedical alloy of hard tissue implants. In addition, the fracture mechanism of the Ti–Ni–Fe alloys is mainly cleavage fracture or quasi-cleavage fracture, supplemented by ductile fracture.

## Figures and Tables

**Figure 1 materials-12-03114-f001:**
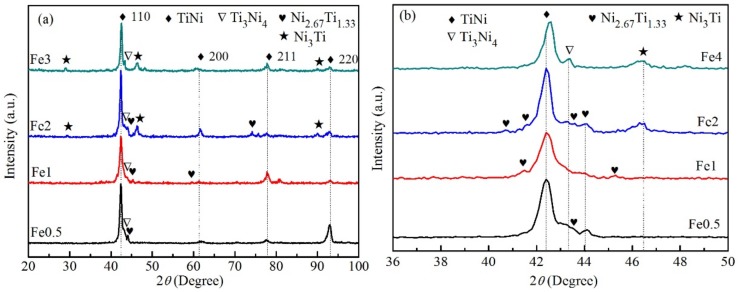
X-ray diffraction patterns of the Ti–Ni–Fe alloys. (**a**) 20°–100°; (**b**) 36°–50°.

**Figure 2 materials-12-03114-f002:**
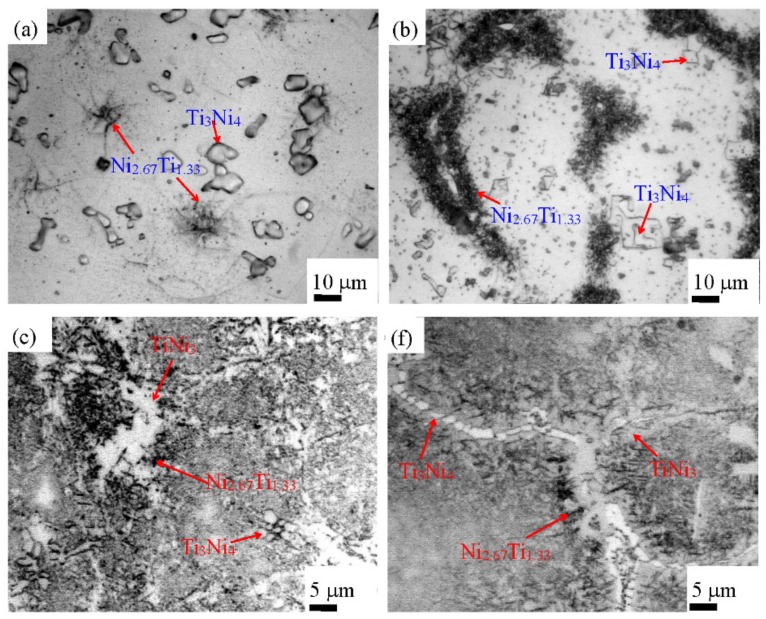
Microstructures of the (**a**) Ti_48.5_Ni_51_Fe_0.5_, (**b**) Ti_48_Ni_51_Fe_1_, (**c**) Ti_47_Ni_51_Fe_2_, (**d**) Ti_45_Ni_51_Fe_4_ alloys.

**Figure 3 materials-12-03114-f003:**
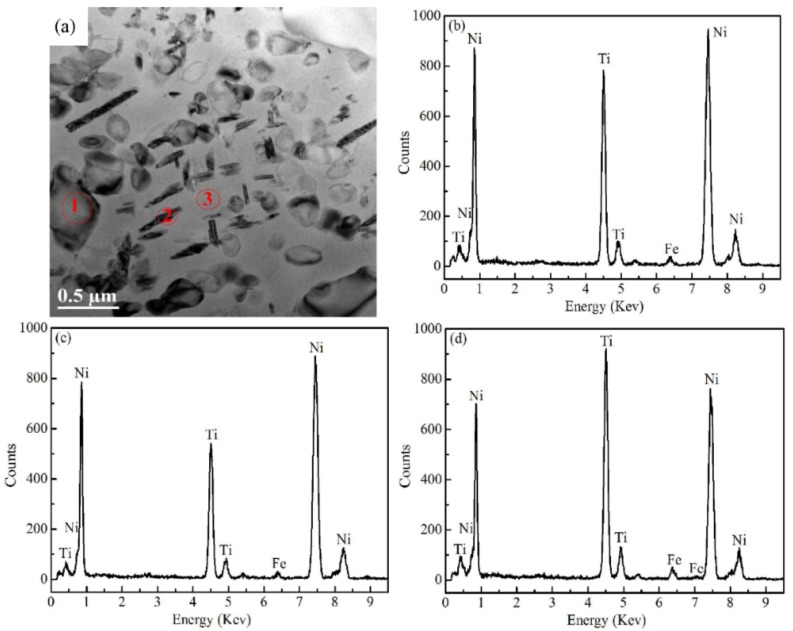
Energy dispersive spectrometer (EDS) of Ti_48.5_Ni_51_Fe_0.5_ alloy. (**a**) TEM bright field image; (**b**) Point 1; (**c**) Point 2; and (**d**) Point 3.

**Figure 4 materials-12-03114-f004:**
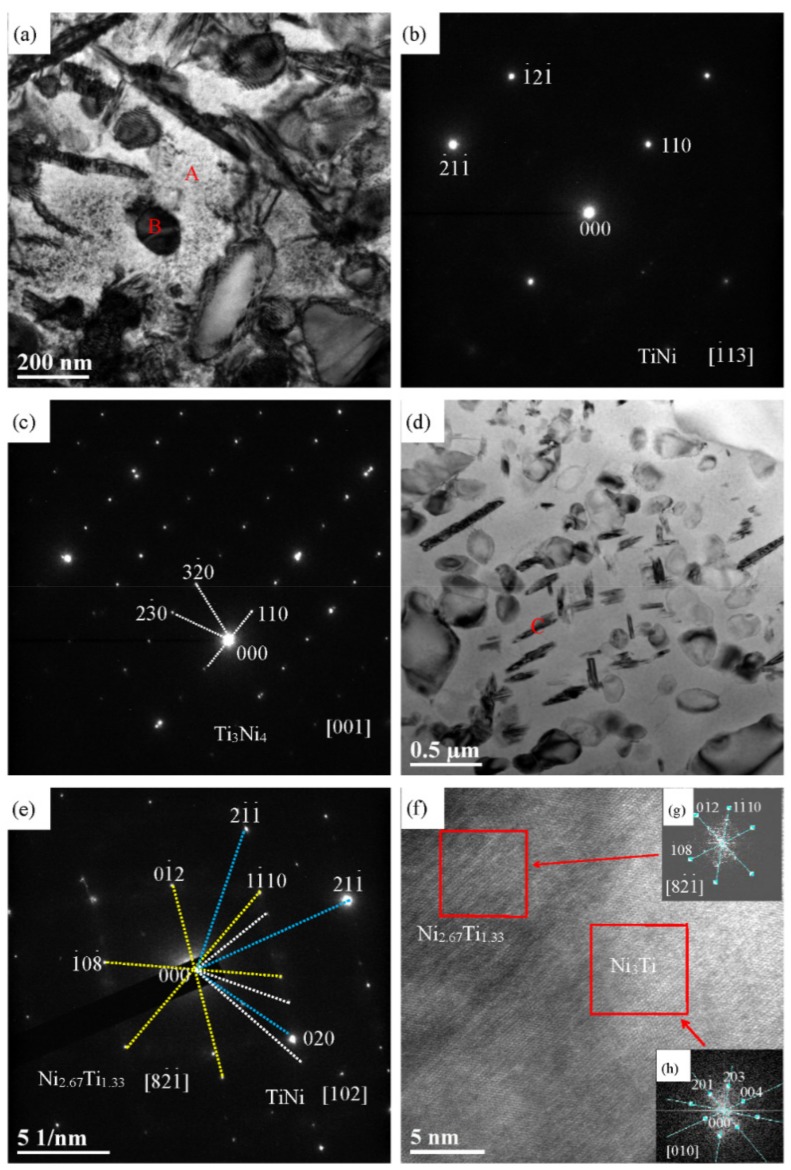
(**a**) and (**d**) TEM bright field images of Ti_48.5_Ni_51_Fe_0.5_ alloy. (**b**) and (**c**) Electron diffraction patterns taken from the areas marked “A” and “B” in (**a**), respectively; (**e**) taken from “C” in (**d**). (**f**) HRTEM image of Ti_48.5_Ni_51_Fe_0.5_ alloy. (**g**) Fourier transform of Ni_2.67_Ti_1.33_ phase; (**h**) Fourier transform of Ni_3_Ti phase.

**Figure 5 materials-12-03114-f005:**
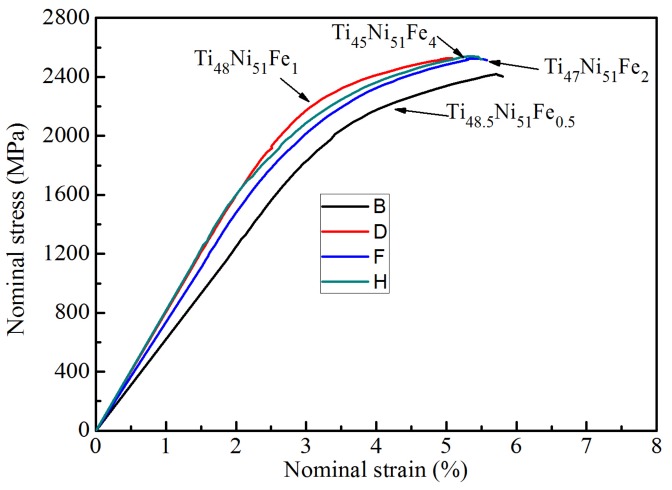
The nominal stress–strain curves of the Ti–Ni–Fe alloys.

**Figure 6 materials-12-03114-f006:**
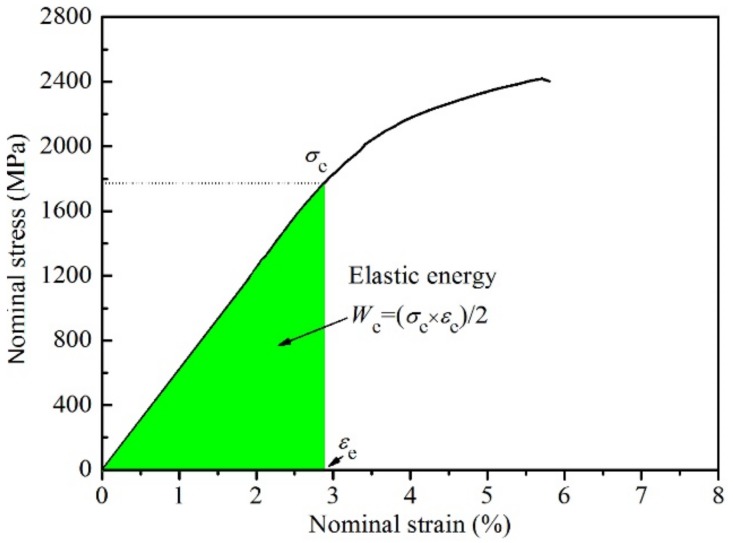
The elastic energy (*W*_e_) of Ti_48.5_Ni_51_Fe_0.5_ alloy.

**Figure 7 materials-12-03114-f007:**
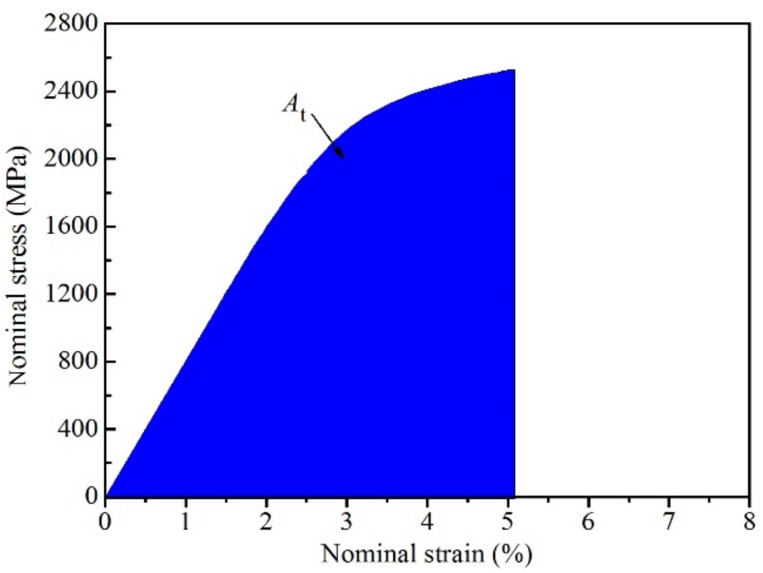
The toughness (*A*_t_) of the Ti_48_Ni_51_Fe_1_ alloy in the compressed stress–strain curve.

**Figure 8 materials-12-03114-f008:**
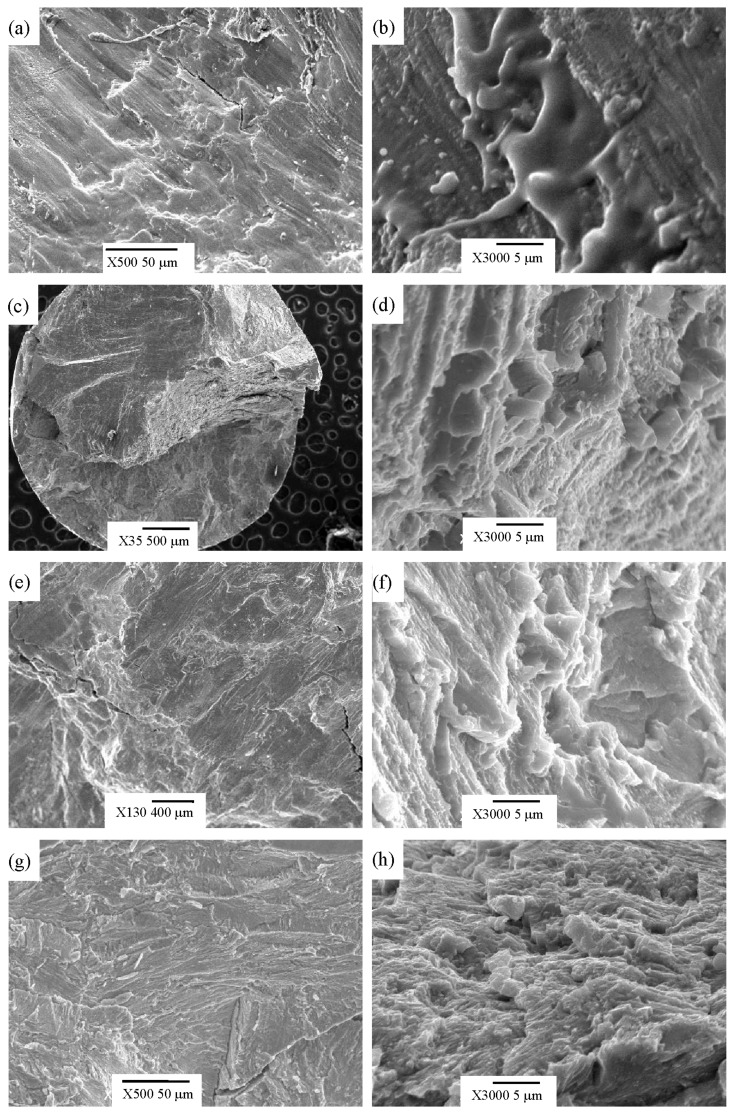
Fracture morphology of as-cast Ti–Ni–Fe alloys compressed at room temperature. (**a**) and (**b**) Ti_48.5_Ni_51_Fe_0.5_, (**c**) and (**d**) Ti_48_Ni_51_Fe_1_, (**e**) and (**f**) Ti_47_Ni_51_Fe_2_, (**g**) and (**h**) Ti_45_Ni_51_Fe_4_ alloys.

**Table 1 materials-12-03114-t001:** Mechanical properties of Ti–Ni–Fe alloys including elastic limit (*σ*_e_), the 0.2% offset yield stress (*σ*_0.2_), plastic strain (*ε*_p_), maximum compression strength (*σ*_b_), elastic modulus (*E*), elastic energy (*W*_e_), toughness values (*A*_t_), approximate volume fraction of phase (*V*).

Alloys	*σ*_e_ MPa	*σ*_0.2_ MPa	*ε*_p_ %	*σ*_b_ MPa	*E* GPa	*W*_e_ × 10^6^ J·m^−3^	*A*_t_ × 10^6^ J·m^−3^	*V* (TiNi) %	*V* (Ti_3_Ni_4_) %	*V* (Ni_2.67_Ti_1.33_) %	*V* (Ni_3_Ti) %
Ti_48.5_Ni_51_Fe_0.5_	1779	2066	2.2	2420	61.8	26.12	89.91	74.24	16.41	9.35	--
Ti_48_Ni_51_Fe_1_	1604	2089	2.87	2529	80.2	16.51	85.23	76.42	10.05	13.53	--
Ti_47_Ni_51_Fe_2_	1566	1945	2.76	2524	73.2	16.69	92.89	68.49	11.81	9.71	9.99
Ti_45_Ni_51_Fe_4_	1463	1865	3.0	2541	81.3	13.51	94.36	74.26	14.77	--	10.97

**Table 2 materials-12-03114-t002:** The percentage content of alloy elements in energy dispersive spectrometer test.

Number	Ti (at.%)	Ni (at.%)	Fe (at.%)
1 2 3	41.3 34.2 50.0	57.9 65.0 48.7	0.9 0.8 1.3

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
