# Peer review of "Effect of Fe Addition on Microstructure and Mechanical Properties of As-cast Ti49Ni51 Alloy"

_materials, 2019, doi:10.3390/ma12193114_

Round 1
Reviewer 1 Report
Review on
«Effect of Fe Addition on Microstructure and Mechanical Properties of as-cast Ti49Ni51 Alloy»
by Peiyou Li, Yuefei Jia, Yongshan Wang, Qing Li, Fanying Meng, Zhirong He
The paper entitled «Effect of Fe Addition on Microstructure and Mechanical Properties of as-cast Ti49Ni51 Alloy» presents a study of Fe addition effect on microstructure and mechanical properties of as-cast Ti49Ni51 alloy for four compositions (Ti48.5Ni51Fe0.5, Ti48Ni51Fe1, Ti47Ni51Fe2, Ti45Ni51Fe4). The originality of this paper comes from considering Ni-rich Ti-Ni-Fe alloys as base ones as they are rarely reported in literature unlike Ti-rich Ti-Ni-Fe alloys. The two compositions containing the less Fe addition shows high yield strength and fracture strength combining to a low elastic modulus and large elastic energy could be interesting for biomedical applications. The topic is of interest to the readers of Materials. The paper is well written and well structured.
In all, this paper makes an interesting contribution to the research field, and the reviewer recommends it for publication on Materials if the following problems are issued.
1. There is no affiliation for two of the authors.
2. In the introduction, line 58, please provide the reference number for Liang et al.
3. In the section 2, line 88, there is a space to remove « to en sure »
4. Concerning the quasi static uniaxial compression tests at room temperature, it is indicated in the section experimental procedure that the tests are performed three times and then that the reported experimental data are the average of three measurements. In section 3.2, figure 5, the nominal stress-strain curves are shown but there is no information about the spread of the data around the mean. According to the reviewer, it should be indicated.
5. In the section 3.2, it is underline that no martensite transformation is observed on the stress-strain curve due to the cristallographic structure of the austenitic phase. In fact, the four compositions tested cannot be considered as a shape memory alloy. Is this not a significant restriction of use of these new alloys ? The main argument used by the authors to enhance these new alloys is high yield strength and fracture strength combining to a low elastic modulus and large elastic energy, which could be interesting for biomedical applications. In fact, implants are generally architectured and/or porous structure, then classical Titanium alloys can be sufficient to reduce the stress shielding.
6. Line 266, probably it should be figure 5 and not figure 1 in the text.
Author Response
Dear editor and reviewer,
The following contents are Editor or Reviewer comments and my answer. The revised contents are labeled by blue words in revised manuscript.
Sincerely yours,
Peiyou Li
Q1. There is no affiliation for two of the authors.
An. Thank for the reviewer proposed the question. We have added affiliation of two authors.
Q2. In the introduction, line 58, please provide the reference number for Liang et al.
An. Thank for the reviewer proposed the question. We have added the reference number for Liang et al.
Q3. In the section 2, line 88, there is a space to remove « to en sure ».
An. Thank for the reviewer proposed the question. In revised manuscript, in line 90, we have remove a space, “To ensure the uniformity of the chemical composition”.
Q4. Concerning the quasi static uniaxial compression tests at room temperature, it is indicated in the section experimental procedure that the tests are performed three times and then that the reported experimental data are the average of three measurements. In section 3.2, figure 5, the nominal stress-strain curves are shown but there is no information about the spread of the data around the mean. According to the reviewer, it should be indicated.
An. Thank for the reviewer proposed the question. In line 265-267, we have added the corresponding contents, “Since the each alloy is tested three times, each alloy has three stress-strain curves; in Fig. 5, the experimental data on the selected stress-strain curves are close to the average of the three measurements.”
Q5. In the section 3.2, it is underline that no martensite transformation is observed on the stress-strain curve due to the cristallographic structure of the austenitic phase. In fact, the four compositions tested cannot be considered as a shape memory alloy. Is this not a significant restriction of use of these new alloys? The main argument used by the authors to enhance these new alloys is high yield strength and fracture strength combining to a low elastic modulus and large elastic energy, which could be interesting for biomedical applications. In fact, implants are generally architectured and/or porous structure, then classical Titanium alloys can be sufficient to reduce the stress shielding.
An. Thank for the reviewer proposed the question. Among the new alloys found, the Ti48.5Ni51Fe0.5 and Ti48Ni51Fe1 alloys with high strength and fracture strength can be used as engineering structural materials because they do not have some properties of shape memory alloys, while the Ti48.5Ni51Fe0.5 alloy with low elastic modulus and high elastic energies can be used as hard tissue implants of biological alloys. In medical implant materials, soft tissue implants have reticular or porous structure, while hard tissue implants can not have martensitic deformation or small volume changes within the temperature range that the human body can withstand. In the manuscript, we mentioned that new alloys can be used in hard tissue implants.
Q6. Line 266, probably it should be figure 5 and not figure 1 in the text.
An. Thank for the reviewer proposed the question. We have revised the correspond content.
Reviewer 2 Report
Dear Authors,
Thanks for your excellent work. You provide novel and interesting data on microstructure and shape memory properties of NiTiFe alloys. The very low Young's modulus along with high strength are very interesting for biomedical applications. Please find below my comments.
1- In the introduction, you talk about recent applications of SMAs such as biomedical and aerospace etc. I think you should also mention the elastocaloric application of SMAs in solid state cooling. This is much more recent and active research topic than traditional applications. Please refer to the articles listed below:
Elastocaloric effect associated with the martensitic transition in shape-memory alloys (physical review letters 2008)
Materials with giant mechanocaloric effects: cooling by strength (Advanced Materials 2017)
Reversible elastocaloric effect at ultra-low temperatures in nanocrystalline shape memory alloys (Acta Materialia 2017)
You may refer to more articles.
I think one of the alloys that you have developed (Ni48.5Ti51Fe0.5) may exhibit interesting elastocaloric effect at low temperatures. Generally, the type of SMAs that exhibit stress-strain curves similar to Ni48.5Ti51Fe0.5, preserve large superelatsicity and elastocaloric effect over a wide temperature range. Please read Ref. 3 suggested above to gain some insight about it.
2- In your introduction, you have referred to articles addressing strain glass materials. Please be careful when borrowing any discussion from those articles since the strain glass topic is highly questionable and is generally referred to as "redundant"... It does not explain anything new.
3- To quantify phase fractions using XRD peaks, you need to make sure that textures are random. Please mention this somewhere.
4- Your TEM work looks very solid. Well done. I could not find any flaw or deficiencies.
5- I am very sorry to say it in this tune, but your discussion of stress-strain data in Fig. 5 does not seem correct to me. This section needs complementary experiments or more detailed discussion.
First of all, by "platform" I think you mean stress plateau. Right? Please correct it.
Second, not seeing a stress plateau does not necessarily mean there is no stress-induced martensitic phase transformation.
As shown by QingPing Sun in a series of systematic papers, stress-induced martensitic phase transformation can occur continuously (second-order) whereas stress-strain curve looks smooth (no plateau). Please read Q.P. Sun articles on nanoscale martensitic phase transformation published in Applied Physics Letters (2013), Acta Materialia (2014 and 2015), and Science China Technological Sciences (2014) to understand the topic.
You may discuss that the stress-strain curves in Fig. 5 might indicate a continuous stress-induced phase transformation. Please borrow their discussion and input it in your paper. I think based on the compositions, your samples undergo smooth stress-induced phase transformation. As I mentioned above, this type of stress-strain data is generally temperature insensitive. Please discuss it in more details.
I think your conclusion that there is no stress-induced martensitic phase transformation because the structure is BCC not B2, is not correct. You need to provide loading and unloading data or caloric measurements.
Kind Regards,
Author Response
Dear editor and reviewer,
The following contents are Editor or Reviewer comments and my answer. The revised contents are labeled by blue words in revised manuscript.
Sincerely yours,
Peiyou Li
Reviewer 2.
Q1- In the introduction, you talk about recent applications of SMAs such as biomedical and aerospace etc. I think you should also mention the elastocaloric application of SMAs in solid state cooling. This is much more recent and active research topic than traditional applications. Please refer to the articles listed below:
Elastocaloric effect associated with the martensitic transition in shape-memory alloys (physical review letters 2008)
Materials with giant mechanocaloric effects: cooling by strength (Advanced Materials 2017)
Reversible elastocaloric effect at ultra-low temperatures in nanocrystalline shape memory alloys (Acta Materialia 2017)
You may refer to more articles.
An. Thank for the reviewer proposed the question. Thank the reviewers for providing some references to let us understand the new application fields of shape memory alloys. In the introduction, we add the elastocaloric application of SMAs in solid state cooling. The added contents are the blue words in the first paragraph of the introduction.
Q2. I think one of the alloys that you have developed (Ni48.5Ti51Fe0.5) may exhibit interesting elastocaloric effect at low temperatures. Generally, the type of SMAs that exhibit stress-strain curves similar to Ni48.5Ti51Fe0.5, preserve large superelatsicity and elastocaloric effect over a wide temperature range. Please read Ref. 3 suggested above to gain some insight about it.
An. Thank for the reviewer proposed the question. We have added the corresponding contents in page 9. The added contents are “In Fig. 5, the maximum elastic strain of Ti48.5Ni51Fe0.5 alloy reaches 2.88%, which is obviously larger than 2% of the structural material, and also larger than the elastic strain of the other three Ti–Ni–Fe alloys. The Ti48.5Ni51Fe0.5 alloy with 2.88% elastic deformation is a kind of smooth hardening superelasticity [4]. This kind of smooth hardening superelasticity is different from the pseudoelasticity of shape memory alloy [4,38,39]. The former originates from the elastic deformation of nanocrystalline microstructures, while the latter is the elastic deformation caused by martensitic transformation with larger particle size [4,38,39]. According to the results reported by Ahadi et al. [4], the deformation of nanocrystalline microstructures indicates that a small amount of B2 phase is transformed into B19¢phase, which also indicates that the current Ti48.5Ni51Fe0.5 alloy may contain a small amount of B2 nanocrystals, and the results need to be further studied in the future. In addition, The transition of nanocrystalline B2 phase to B19¢ phase can produce elastocaloric effect in a wide temperature range [4]. Therefore, the elastocaloric effect of Ti48.5Ni51Fe0.5 alloy needs to be further studied in the future.”
Q3. In your introduction, you have referred to articles addressing strain glass materials. Please be careful when borrowing any discussion from those articles since the strain glass topic is highly questionable and is generally referred to as "redundant"... It does not explain anything new.
An. Thank for the reviewer proposed the question. As a researcher who began to study Ti-Ni shape memory alloys, strain glass materials are really unknown. In the introduction, I deleted the corresponding contents and references. The strained glass is not used for discussion in the later part.
Q4. To quantify phase fractions using XRD peaks, you need to make sure that textures are random. Please mention this somewhere.
An. Thank for the reviewer proposed the question. We have added the corresponding contents. The added contents are “Since the four alloys are as-cast alloys, the orientation of each grain is almost completely disorder, that is to say, the orientation probability of the grain is the same, or textures are random. Therefore, the intensity of XRD diffraction peak can be used to quantify phase fraction. ”
Q5. Your TEM work looks very solid. Well done. I could not find any flaw or deficiencies.
I am very sorry to say it in this tune, but your discussion of stress-strain data in Fig. 5 does not seem correct to me. This section needs complementary experiments or more detailed discussion.
First of all, by "platform" I think you mean stress plateau. Right? Please correct it.
Second, not seeing a stress plateau does not necessarily mean there is no stress-induced martensitic phase transformation.
As shown by QingPing Sun in a series of systematic papers, stress-induced martensitic phase transformation can occur continuously (second-order) whereas stress-strain curve looks smooth (no plateau). Please read Q.P. Sun articles on nanoscale martensitic phase transformation published in Applied Physics Letters (2013), Acta Materialia (2014 and 2015), and Science China Technological Sciences (2014) to understand the topic.
You may discuss that the stress-strain curves in Fig. 5 might indicate a continuous stress-induced phase transformation. Please borrow their discussion and input it in your paper. I think based on the compositions, your samples undergo smooth stress-induced phase transformation. As I mentioned above, this type of stress-strain data is generally temperature insensitive. Please discuss it in more details.
An. Thank for the reviewer proposed the question. The word “platform” is “plateau”, we have revised it. By reading the references provided by reviewers, we have revised the discussion of stress-strain data. The blue font is the modified content in page 9.
Q5. I think your conclusion that there is no stress-induced martensitic phase transformation because the structure is BCC not B2, is not correct. You need to provide loading and unloading data or caloric measurements.
An. Thank for the reviewer proposed the question. In order to express the structure of the alloy more accurately, we have added a word “mainly” in abstract, 3.1, and conclusion.
For four alloys, the matrix phase is BCC structure, not B2 matrix structure, based on four aspects of testing, stress-strain curve, TEM, fracture morphology, and DSC experiments. On the stress-strain curve, because there is no macro martensite transformation platform, according to the references provided by the reviewers, we know that there is no large particle size B2 matrix phase in the matrix, which may contain a small amount of B2 nanocrystals. Stress-induced transformation occurs under stress. According to TEM observation, there is no large size B2 phase, possibly nano-sized B2 phase, which has not been found under the current experimental conditions. Fracture morphology can best explain the existence of a large number of TiNi phases with body-centered cubic structure in the alloy matrix, because large area cleavage fracture can only occur in the body-centered cubic and dense hexagonal. If the alloy contains a large amount of B2 phase, the fracture surface will show a large area of ductile fracture morphology. DSC curves were also tested at temperatures ranging from - 100 to 150 degrees. There were no phase transition peaks in the four alloys during heating and cooling, so they were not in the manuscript. In fact, for another alloy system, the stress-strain curves of Ti49Ni51-xFex (x = 0.5, 1, 2, 3 at.%) alloys have no martensitic transformation platforms. DSC measurements show that except for Fe1 alloy with a very small transformation peak, there are no transformation peaks in the other three alloys. The manuscript of Ti49Ni51-xFex (x = 0.5, 1, 2, 3 at.%) alloys has been submitted to other magazines. In inference [16], the transformation peak of Ti49Ni49Fe2 alloy is not found in the DSC curve.